# Companion Animal Fostering as Health Promotion: A Literature Review

**DOI:** 10.3390/ijerph20136199

**Published:** 2023-06-21

**Authors:** Christine Roseveare, Mary Breheny, Juliana Mansvelt, Linda Murray, Marg Wilkie, M. Carolyn Gates

**Affiliations:** 1School of Health Sciences, Massey University, Wellington 6140, New Zealand; l.murray1@massey.ac.nz; 2School of Health, Victoria University of Wellington, Wellington 6012, New Zealand; mary.breheny@vuw.ac.nz; 3School of People, Environment and Planning, Massey University, Palmerston North 4410, New Zealand; j.r.mansvelt@massey.ac.nz; 4Research Centre for Hauora and Health, Massey University, Wellington 6140, New Zealand; 5School of Veterinary Science, Massey University, Palmerston North 4410, New Zealand; c.gates@massey.ac.nz

**Keywords:** fostering, companion animals, pets, ageing, human–animal interaction, human–animal bond, health promotion, Te Whare Tapa Whā, Ottawa Charter

## Abstract

There is growing interest in the health-promoting potential of human-companion animal relationships from a broad public health perspective while acknowledging barriers to ownership, particularly for older adults. Companion animal fostering is an alternative to pet ownership that aligns with the Ottawa Charter health promotion principle that caring for others in everyday settings promotes health. This narrative review of the literature on companion animal fostering draws on Te Whare Tapa Whā (the four-sided house), an indigenous model of health that is influential in Aotearoa/New Zealand, and the Ottawa Charter. We found that companion animal fostering can be considered health-promoting for human and non-human animals, using a broad and multidimensional understanding of health. As well as improving the long-term outcomes for homeless animals, companion animal fostering has the potential to promote the health of the individuals, families, and communities who provide foster homes. Our review highlights the importance of health promoters considering the reciprocal relationship between human and animal health. Future research should explore different aspects of human and non-human health, perspectives of different types of fosterers in different settings and communities, barriers to fostering, and methods that explore the role of caring for a wider range of companion animals in creating and sustaining wellbeing.

## 1. Introduction

‘The welfare of people and other animals is closely linked’ [1].

According to the Ottawa Charter for Health Promotion [2], health is created and sustained in the settings where we live, work, and play, with interdependent relationships [3] and our care for others playing an important role [4]. Caring for others is traditionally framed in the context of human relationships, but it is also important to recognise our interdependent relationships with non-human animals [5,6,7]. Increasingly, we are in a world of multi-species households where humans share a home with one or more non-human animal companions and may think of those animals as members of the family [8,9]. In recent years, interest in our interdependence and relationships with companion animals has grown in the social sciences, psychology, arts, and humanities [10,11,12,13,14]. Human–animal interaction studies is a broad and growing interdisciplinary field [15] and is one in which health promotion as a discipline is starting to engage [16]. Recent research has considered the health-promoting potential of human–companion animal relationships from a broad public health perspective, including the challenges companion animals and their guardians may face [7,17,18].

The growing body of literature on the health impact of relationships between humans and companion animals has shown some promising but overall inconsistent empirical evidence of the mental, physical, emotional, and social benefits for humans [19]. One explanation for the variability in the health outcomes is the nature and intensity of the caregiving relationship [20], including the frequency, proportion, and nature of the time spent with an animal companion [21]. Fleeting contact may provide little health benefits [22], but direct involvement in caregiving may have a significant effect [23] and provide a sense of purpose and meaning that contributes to wellbeing [24,25,26,27].

Caring for and having a meaningful connection with companion animals does not necessarily require ownership [28,29,30,31]. An increasing number of rescue organisations run companion animal foster programmes where stray or surrendered animals are placed in temporary homes with human fosterers at a time of transition until they are ready to be adopted [32] or in some cases until they can be reunited with their original human guardian. Animals may be fostered because they are too young to rehome, would benefit from socialisation, medical, or behavioural rehabilitation, [28,33], because of a lack of capacity in an animal shelter, or in response to an emergency event [34,35]. Community fosterers live, work, and play with foster animals for periods ranging from a few days to several months or longer. Fostering may also involve a longer-term arrangement designed to ensure permanent homes for harder-to-rehome animals. Fostering is a relationship with caring for others at its heart.

Much of the literature about fostering focuses on the benefits of animal welfare, but fostering may also be health-promoting for humans. The literature on the health benefits of companion animals has focused mostly on pet ownership situations, where individuals or families form permanent long-term relationships with their animals, [36] or animal-assisted interventions where individuals often have transient contact with animals that are owned or cared for by others [37,38]. There is limited literature on the human health impacts of temporary guardianship relationships with animals that involve active caregiving and a potentially close relationship without ownership [39]. For older adults who are facing challenges with life transitions and ageing, fostering may be an alternative to permanent pet ownership. Some older people struggle with the costs of animal ownership, express concerns about dying before a pet, or live in situations that restrict pets (e.g., aged care facilities) [19,27,28,29,30,31,40]. Consequently, examining alternatives to ownership that include the caregiving of companion animals by older people may be valuable. Fostering may also allow for more reciprocal and meaningful relationships than are possible in animal-assisted interventions, such as visiting or therapy programmes [7,22]. Fostering has the potential to contribute to the capability and resilience of older people [28] whilst recognising the reality of the challenges associated with ageing [41].

This review contributes to an emerging body of work seeking to integrate health promotion and animal welfare perspectives [5,6,7,17,42]. The Ottawa Charter provides three key foundations for this review. The Ottawa Charter acknowledges health as broad and interconnected, draws attention to matters of equity, and sees relationships of care as fundamental to health. The charter also provides for an inclusive definition of care if care for others is not limited to human relationships [3,4]. Relationships with companion animals may contribute to wellness by providing a sense of purpose [25], particularly for older people. We ask the question ‘Can fostering be considered health-promoting for human and non-human animals?’ The objectives of this review are to summarise the published research on companion animal fostering as health-promoting for non-human animals and humans, with particular reference to older people, and identify important gaps for future research on fostering.

## 2. Health Promoting Aspects of Fostering for Humans and Animals

To determine if fostering is health-promoting, we consider what is known about its potential to create or support health and promote equity. Our understanding is informed by the Ottawa Charter for Health Promotion (see Table 1). As the Ottawa Charter does not specifically define health, we draw on Te Whare Tapa Whā (Table 1), an Indigenous model of health that is influential in Aotearoa/New Zealand and consistent with the broad concept of health expressed in the Ottawa Charter and the subsequent Bangkok Charter [43]. Using this framework to organize and interpret the literature allows us to see aspects of fostering and health that we might otherwise overlook.

Te Whare Whā describes health as having four inter-related dimensions: physical, mental, social, and spiritual [44]. We then turn to consider the equity aspects of fostering before highlighting important areas for future research.

**Table 1 ijerph-20-06199-t001:** Health-promoting potential, the Ottawa Charter, and Te Whare Tapa Whā.

The Ottawa Charter The Ottawa Charter, a guiding document for health promotion [2], argues that care for others and reciprocal notions of care are fundamental to what creates and sustains our health [3,4]. Things are health-promoting that create or support the health of individuals, families, and communities. Health is seen as a resource for everyday life, not an endpoint. Health is created and sustained through interdependent relationships with others in the settings where we live, work, and play, and it is determined through a broad range of factors. Not everyone has equal access to the resources that support health. The Ottawa Charter has a broad view of health, which is more than the absence of disease.
Te Whare Tapa Whā In Aotearoa/New Zealand, the Indigenous Māori people have a model of health—Te Whare Tapa Whā—created by respected Māori health leader Professor Sir Mason Durie [45], which is consistent with the broad concept of health represented in the Ottawa Charter [44]. Te Whare Tapa Whā describes health as a balance of four inter-related dimensions: taha tinana (physical), taha hinengaro (mental and emotional), taha whānau (extended family and social relationships), and taha wairua (spirituality). This model sees the wellbeing of people and their environments as inextricably linked. Companion animal fostering potentially engages all of the dimensions of wellbeing from this holistic perspective.

### 2.1. Physical Health (Taha Tinana) and Emotional Health (Taha Hinengaro)

The dimensions of taha tinana and taha hinengaro in Te Whare Tapa Whā acknowledge the physical and emotional dimensions of health and see these as foundational to wellbeing. The wellbeing of human and non-human animals is interconnected [46] and both are worthy of consideration.

Fostering programmes may benefit animals physically by helping to address capacity issues in animal shelters, decreasing the frequency of pathogen transmission, and reducing the risk of euthanasia that surrendered or stray animals may face [47]. Foster care has been associated with improved odds of live release for dogs in a study in the USA [33], and foster programmes are credited by rescue organisations and shelter staff as effective in reducing euthanasia and increasing adoption rates [48,49,50,51]. Focusing specifically on the impact of foster care on survival rates for cats, Kerr et al. found that the use of foster care contributed to improved outcomes, including increased live release rates and reduced rates of euthanasia. Foster care was particularly valuable for improving outcomes for older cats [52].

Animal shelters can be stressful environments for animals. Fostering in the everyday setting of the home provides a respite from the stress of shelter life, opportunities for play and other positive interactions with humans, and may increase the likelihood of successful adoption [28,47,53,54]. Although studies exploring the impact of fostering on the health of dogs or cats are still rare, Gunter et al. [55] demonstrated that short-term stays in a foster home reduced stress in dogs, and Vitale et al. [56] showed that very short-term foster stays neither increased nor decreased stress levels for cats. Graham’s research exploring the impact of fostering on fear in kittens suggested that the outcomes depend on a range of factors, including the personality, experience, and practices of the fosterer and the variability within the kittens. Not all fosterers used recommended socialization techniques, and some fostered kittens may miss out on experiences that are important for managing fear in later life. This suggests that training and support for fosterers is important to ensure best practices when working with fearful kittens [57]. Studies considering other potential damaging impacts of a fostering experience on cats found no association between fostering and later aggression in cats [58] or poorer growth in kittens [59]. Overall, research on the benefits of fostering for companion animal health is promising [53].

Fostering may have benefits for the fosterers as well. Two recent interventions in the United States have focused on connecting older people living alone in the community with a foster animal, with the objective of improving physical and emotional health. The first small-scale pilot intervention involved older veterans with emotional and physical health challenges fostering a dog for a two-month period [40], and the second involved older people who lived alone fostering a cat for four months [60,61]. The first intervention reported modest but positive benefits regarding physical activity and quality of life, and the second study, which is still in progress, identified a reduction in loneliness. Both these studies aimed to enable the eventual adoption of the foster animal by the foster guardian, which provides a route to animal ownership.

In contrast, fosterers who volunteer with animal rescue or service organisations often foster on an ongoing basis, although some may foster only once [55,62]. Volunteer fosterers may experience both emotional health benefits and challenges. Recent research by Reese found that nearly all the fosterers surveyed agreed that fostering dogs added to their happiness, and most felt that interacting with their foster animals helped them stay healthy [39]. Fosterers describe a sense of connection with animals in their care and satisfaction and elation when foster animals thrive and are adopted. In addition to this, fosterers may also experience fostering as emotionally demanding and experience compassion fatigue, burnout, and grief [62,63,64,65].

Despite being motivated by a love of animals, some research suggests that to continue caring for animals on a temporary basis, fosterers may need to manage their fostering relationship carefully [63]. They may need to interact without becoming too attached or recognise and accept attachment but be prepared to relinquish this attachment [63,66]. Reese et al. explored aspects of fosterer attachment and found high levels of attachment for fostered animals among fosterers. Rather than being associated with thoughts of quitting fostering, attachment to animals was associated with a willingness to continue and mitigated some of the stress associated with fostering high-needs animals [39]. Dewitt found that while it was real, the sense of loss experienced when the foster animal was adopted or returned did not necessarily prevent fosterers from fostering again [65]. This suggests that the nature of the relationship might include recognition of meaningful but temporary attachment.

### 2.2. Family and Social Relationships (Taha Whānau)

The dimension of taha whānau in Te Whare Tapa Whā recognizes the importance of relationships with family and community regarding health. The health of the individual is not separated from the health of the collective. In Te Ao Māori (the Māori worldview), the term whānau (extended family) can include those that are whāngai (fostered) [42].

Descriptive studies of volunteer fosterers show that some fosterers live with partners, children, or other animals, meaning some foster animals may be joining multi-species families. It is important to consider that fostering may have negative impacts on other household animals through disease transmission [67] or stress from the frequent addition of other foster animals to their environment [68]. Recognition of these extended human and non-human household relationships may contribute to a more nuanced understanding of fostering. Few studies explore the impact of fostering on the health or welfare of the wider family, consider the contribution of the family to the fostering relationship, or view the foster animal as part of the family.

The main aspect of relationships beyond the household explored in the fostering literature are the social ties and sense of community fosterers experience with other fosterers. Roemer [63] and Daily [64] both comment on the value fosterers placed on being part of a supportive community, and DeWitt [65] and Graham [57] describe how support from other fosterers was highly valued.

The potential for collective rather than individual fostering is an area for development. Armitt et al. [69] describe a proposed intervention where the staff and residents in an aged care setting share the care of a foster cat rather than relying on individual responsibility for care. McDonald acknowledged that shared informal care for cats may already occur in communities and suggested research on how agencies could support such beneficial informal fostering arrangements [47].

### 2.3. Spiritual Wellbeing (Taha Wairua)

Taha wairua, or spirituality, provides a lens through which to consider aspects of meaning, purpose, and values and questions of life and death as dimensions of health [70]. To date, these aspects are only considered briefly in the fostering literature. Barret and Patlamazoglou [71] in their conference abstract of an interview study, found that fosterers described the meaning of fostering animals and their experiences of animal death as important to them, while Daily’s research with dog fosterers [64] found that a wish to ‘contribute’ was one of the most common reasons for choosing to foster, suggesting a sense of purpose may play a role in fostering.

Using Te Whare Tapa Whā as a framework enables a broad view of health, includes dimensions relevant to companion animal fostering that may otherwise be overlooked, and alerts us that these dimensions may be interconnected. Aspects of community, emotional, and spiritual health are apparent in Katja Guenther’s research [72]. Guenther describes how shelter volunteers work to ensure animals who are euthanised do not die without mention and that the community marks their passing. Fostering is built on meaningful relationships. Experiences of grief and loss for foster animals and the recognition of fostering as a contribution to the community are complicated and multifaceted. This aligns with Te Whare Tapa Whā, which acknowledges multiple dimensions of health that cannot be separated.

### 2.4. Equity and Diversity

Health promotion is built on the recognition of processes of inequity and commitment to promoting equitable arrangements for health. Fostering programmes potentially reduce financial barriers to caring for animals, as some rescue organisations supply fosterers with the resources to care for fostered animals, including food, bedding, litter, toys, and vet care. However, there may be other barriers to fostering, including a lack of transport, housing policies that do not allow animals [7,18], or a lack of engagement with historically marginalised populations [73,74], contributing to a lack of diversity in the fostering workforce.

Studies of volunteer fosterers identify them as predominantly female [54,57,63,64,75] under 65 years, and when ethnicity is reported, they are identified as members of ethnic majorities. This aligns with the findings from research on animal rescue volunteers as a whole [76]. Most of the information on fosterers comes from North American fosterers, but a study of Brazilian fosterers [75] had a similar demographic profile.

Recent research on the animal rescue workforce, including animal fosterers, has begun to consider diversity and argued to address the lack of diversity [47,53,73,77,78]. Roberts and colleagues’ geographic analysis of Canadian shelter data [78] found that based on the deprivation profiles of the area they lived in, fosterers were more broadly representative of the community than might be expected from previous studies of animal rescue volunteers. Nevertheless, they argued that rescue organisations need to review and adapt their strategies to effectively engage under-represented communities in fostering programmes.

Similarly, McDonald et al. argue for the importance of promoting diversity and equity in fostering programmes. Their research with predominantly Hispanic/Latinx adults from areas with a high intake of kittens to animal shelters found that community members were largely unaware of the organisation’s fostering programmes, although there was informal care for homeless animals occurring that might be supported more formally [47]. There was no research identified on Indigenous communities with fostering programmes. The overall lack of diversity in the fostering literature suggests a gap in engagement with diverse communities.

## 3. Discussion: Gaps and Opportunities

Companion animal fostering can be considered a health-promoting activity for human and non-human animals if our definition of health is broad enough. This aligns with the Ottawa Charter, which sees health as a resource for everyday living, and Indigenous models of health, such as Te Whare Tapa Whā, which state that health is multidimensional and interconnected. Understandings of health that acknowledge emotional and spiritual wellbeing and the interrelatedness of human and non-human communities allow us to go beyond the physiological needs of animals and their human guardians. Using such understandings, we can conceptualise health as enhanced through relationships of care, which might include the challenging and difficult experiences of fostering as well as the joyful and rewarding experiences.

Fostering is not without cost or risk; some people only foster once, whereas others continue to foster and report that fostering supports their health and wellbeing. For these fosterers, their sense of attachment and commitment to the animals they care for is a resource that sustains them, aligning with the Ottawa Charter claim that care for others is important for health. Fostering also provides opportunities to develop and sustain social ties and a sense of purpose.

Much of the research on fostering has been cross-sectional, captures data at a single point in time, ref. [60] and reflects the perspectives of women from majority groups. Sayers and Forrest note that one reason for this may be recruitment techniques that rely on snowball sampling using social marketing [42]. We found only two studies that included in-depth interviews with fosterers [62,63] and none that explored Indigenous perspectives. Cross-sectional methods and quantitative measures provide some useful insights into aspects of fostering but are limited in their ability to capture the mechanisms by which aspects of the fostering relationship promote fosterer health, the interdependent nature of the fostering relationship, or its diversity. Research examining settings beyond North America, other genders, and fostering relationships over time and in greater depth would strengthen the research base.

The focus of most fostering research to date on the benefits for animals has been on the potential to increase survival rates and support the emotional health of dogs. Only one study considered the impact of fostering on both the fostered animals and the fosterer [57]. Given the interdependent nature of companion animal relationships, it would be valuable to consider the impacts of the relationship on both parties. Gaps and areas to explore further (see Table 2) include exploring the relationship between companion animal fostering and different aspects of human and non-human health, the perspectives of different types of fosterers in different settings and communities, barriers to fostering, and measures and methods that explore the role of caring for a wider range of companion animals in creating and sustaining wellbeing.

### 3.1. Fostering as Health-Promoting for Older People

The Ottawa Charter seeks to enable all people to achieve their fullest health potential and Te Whare Tapa Whā sees physical, emotional, mental, family, and spiritual health as inextricably linked to social and community life and physical and societal environments. Relationships of care are fundamental to health, and reciprocal relationships are particularly important in later life [79]. Older people in particular may need support to realise their aspirations as their physical capacities alter. Health interventions using companion animal fostering to reduce loneliness, alleviate ill-health, and reduce barriers to adopting an animal companion for older people are starting to emerge. However, these studies tend to view older people as targets for intervention rather than capable and contributing members of society. There are few studies on the experiences and benefits to older people of fostering animals as part of their everyday lives.

Further research on companion animal fostering offers the chance to improve our understanding of how caring for others may be a mechanism to promote health, particularly in later life. A sense of purpose and connection beyond oneself is important in later life, and caregiving and guardianship include aspects of taha wairua or spiritual health that may be particularly useful to explore with older people.

### 3.2. Reconceiving the Ottawa Charter to Encompass Human–Animal Relationships

To consider the health promotion potential of companion animal fostering, our review incorporated three important concepts from the Ottawa Charter: (1) a broad understanding of health, (2) that caring for others is part of what creates and maintains health, and (3) equity is fundamental to health, which were all reconceived to include non-human animals.

Our review supports a conceptualisation of health promotion that includes relationships with non-human animals. The emerging evidence that companion animal fostering, which fundamentally revolves around the act of caring, can bring potential health benefits to both animals and humans provides one example of the importance of human–animal relationships. We support reconceiving the Ottawa Charter to encompass human–animal relationships and propose that future versions of Health Promotion Charters should reconceive the concepts of health, caring for others, and equity to include both human and non-human animals.

### 3.3. Limitations

We have used Te Whare Tapa Whā framework as a way to consider aspects of the health impact of fostering from a health promotion perspective. The application of Te Whare Tapa Whā recognises the unique context of health in the setting where the authors live—Aotearoa/New Zealand. A potential limitation of our approach is that the alignment between Te Whare Tapa Whā and the understanding of health represented in the Ottawa Charter may not resonate with health promoters outside Aotearoa/New Zealand.

The framework is consistent with the broad concept of health represented in the Ottawa Charter and highlights aspects of health, such as spirituality and family, that might otherwise be missed. Our exploration of these aspects is initial and does not claim to represent a full understanding of these concepts or adequately convey the full significance that the original Māori terms may represent for wellbeing [80]. We encourage other researchers to explore these issues further in different contexts. In addition, framing our discussion around the four domains separately was useful for clarity but does not fully represent an aspect of Te Whare Tapa Whā, which emphasises the importance of balance between the four domains.

## 4. Conclusions

Companion animal fostering has the potential to promote and sustain the health of humans, the animals they care for, and the families and communities they are part of. Alignments with the Ottawa Charter’s broad perspective of health, concern for equity, and claims that health is created by caring for others are apparent in the fostering literature. Considering these alignments offers opportunities to theorise human–animal relationships in more nuanced ways and develop health-promoting interventions that benefit both humans and companion animals. Fostering takes place in communities. These communities have diverse perspectives on the best way to care for animals. They also experience inequities regarding access to the resources that shape health and contribute to both human and animal homelessness. Future research should incorporate such perspectives, recognise the interdependence of humans and non-human animals, multi-species families and communities, and bring a broad perspective to this potentially health-promoting activity.

## Figures and Tables

**Table 2 ijerph-20-06199-t002:** Eight areas to explore in future research on companion animal fostering.

1. The impact of fostering periods of several weeks or longer on companion animals
2. Fostering among currently under-represented groups, including Indigenous people and other marginalized communities, older people, and men
3. Potential barriers to fostering including emotional barriers, such as fear of the pain of loss, and economic barriers, such as cost and transportation
4. Fostering programmes in different community-based and residential settings, including the potential for communal or team fostering
5. Exploring the interdependent nature of the fostering relationship considering the impacts on both the foster guardian and the companion animal
6. The extent to which fostered companion animals are thought of as part of the family and the implications of fostering on the health of the family
7. The relationship between fostering and a sense of meaning and purpose
8. The nature of attachment grief and the loss of fostered animals, either through death during fostering or after return to the shelter as well as ‘loss’ to adoption.

## Data Availability

No new data were created or analysed in this study. Data sharing is not applicable to this article.

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
