# Peer review of "Companion Animal Fostering as Health Promotion: A Literature Review"

_ijerph, 2023, doi:10.3390/ijerph20136199_

Round 1

Reviewer 1 Report

This is a very interesting narrative review of how animal fostering could interact with human and non-human animal welfare. My main comments are overarching, about how the manuscript is framed.

It isn’t really clear how the Ottawa doc used except for the section on equity. Otherwise, it is mostly used as background. Please adjust the introduction and other spots to be clear how the two documents are being used and where there is overlap.

Overall, the negatives of pet ownership or fostering are only lightly touched on. Please provide a more balanced perspective throughout the manuscript. I’ve highlighted a few specifics below. Zoonoses, loss of pet, only have one sentence on this.

Related to the above, there is a serious bias in who is studied in pet ownership research as well as who owns pets. And mostly it highlights the benefits. Yet we know there are challenges. Please include a bit more on this issue in the text. See for example, Saunders et al. Plos One 2017 “Exploring the differences between pet and non-pet owners: Implications for human-animal interaction research and policy”.

Line 160: or that people who are concerned about that loss don’t foster to begin with?

Paragraph starting line 162: is this meant as a summary of above? Anything to add from Graham’s PhD?

Line 253: statement “but fosterers continue”. Only some do…and I don’t think we have good data on who doesn’t and why.

Line 263: other genders. For pet ownership, it is primarily women who identify as owners. Is that related and partly causing the challenges of mostly women fostering?

Line 266: why only dogs here?

Line 278: is “Relations” the accurate word or is this relationships? Just checking.

Line 291: also, loss as many older people are losing their friends and family. Is losing a foster pet (to adoption even) too much? And what about other barriers like transportation and shelters that don’t pay for supplies?

Table 2:

1. What do the authors consider to be longer term? Please include it in the table.

3. What are the different settings? Please be more specific in the table.

5. Does this include other non-human animals? Please be specific here.

7. This would include both death of the foster, either during fostering or after return to the shelter as well as “loss” to adoption. Please clarify in the table.

Conclusion seems to focus primarily on older adults. But the rest of the manuscript is much broader. Please edit the text.

Ref 50: electronic thesis reference info needed.

Inconsistent use of oxford comma. Please add throughout.

Reviewer 2 Report

This article is a review on the topic of the health benefits of companion animal fostering. It has two obvious merits in my view: the first is to employ a, interesting and multi-dimensional theoretical framework to define the concept of health, and the second is to offer suggestions to direct future research. In this respect, it offers what a review should offer. However, I have some perplexities about the study that inclines me to think that this article deserves a further round of revisions.
The first perplexity I have concerns the level of detail employed in describing and adopting the theoretical frame employed to define the concept of health. It is certainly a suggestive and potentially powerful frame, but at the same time it is handled somewhat vaguely by the authors.
This is especially evident in the treatment of the third and fourth dimensions of health. While, for example, it is plausible that there is a connection between the health of the individual and the family and community of which  she is a part, the description of this connection that is provided is unclear and scarcely detailed. Moreover, family and community constitute two quite distinct social entities and can have quite different influences on the individual (which, by the way, is evident in the text, where the different effect of fostering concerning the family and social dimensions is emphasized). From this point of view, putting them together in a single category is a weak move. Finally, different cultures may follow quite different social and family models. To talk about "family" and "society" without clear cultural and historical references is rather abstract.
Similar remarks can also be made regarding the fourth dimension, the spiritual. Here, too, the proposed definition is generic and abstract from any concrete cultural and historical considerations. All human beings need a family, society, and spiritual life: but the actual content of these institutions, despite a common core, varies widely on historical and cultural grounds that are poorly accounted for.
Despite this limited view of the concrete social variety, the contribution rightly raises the issue of differential access to the benefits of fostering, which is potentially distributed on an unequal basis along the dimensions of gender, age, ethnicity, etc. Three issues need to be raised here in this regard. The first is regarding the use of the term "European," which does not, as done in the text, indicate any particular ethnicity, but refer to the people living in a particular continent. The second is regarding the use of the term "Latinx," which, although done with absolutely good intentions, may be controversial. A recent Gallup survey indicates that it is used by only a minimal fraction of people part of the communities it refers to. I believe that beyond our good intentions, it is the communities who must decide what name to use to refer to themselves. I would therefore discourage the use of the term "Latinx" if the authors are not part of those communities to which the term would like to refer. The third regards the calls for more study with a focus on indigenous groups, which is a good thing, but which should perhaps be accompanied by a call also to focus on other marginalized ethnic communities (in general on all marginalized ethnic communities) whose identity may vary from place to place, that are not necessarily indigenous, and that, as it appears from the authors' discussion of the topic in the text, seems little considered in studies of this type.
Finally, I think the discussion section is too short compared to the rest of the article and needs to be expanded. In general, the value of such an article lies on the one hand in the methodology applied, and in the stimuli it manages to provide. The methodology here is interesting, notwithstanding those elements of vagueness previously indicated. On the other hand, the discussion remains underdeveloped in my opinion and needs to be expanded to do justice to the topic addressed.

Round 2

Reviewer 2 Report

I am fine with the present version of the manuscript. I would like to express sincere gratitude to the authors for the opportunity of reading their work.